# A One-Year Prospective Study of Work-Related Mental Health in the Intensivists of a COVID-19 Hub Hospital

**DOI:** 10.3390/ijerph18189888

**Published:** 2021-09-20

**Authors:** Nicola Magnavita, Paolo Maurizio Soave, Massimo Antonelli

**Affiliations:** 1Postgraduate School of Occupational Medicine, Università Cattolica del Sacro Cuore, 00168 Rome, Italy; paolomaurizio.soave@policlinicogemelli.it; 2Department of Woman/Child & Public Health, Fondazione Policlinico Universitario Agostino Gemelli IRCCS, 00168 Rome, Italy; 3Department of Emergency, Anesthesiology and Resuscitation Sciences, Fondazione Policlinico Universitario Agostino Gemelli IRCCS, 00168 Rome, Italy; massimo.antonelli@unicatt.it

**Keywords:** longitudinal study, emergency, infectious disease, organisational justice, stress, loneliness, compassion fatigue, meditation, prayer, insomnia, mental health, anaesthetists, occupational health

## Abstract

The COVID-19 pandemic has severely tested the physical and mental health of health care workers (HCWs). The various stages of the epidemic have posed different problems; consequently, only a prospective study can effectively describe the changes in the workers’ health. This repeated cross-sectional study is based on a one-year investigation (spring 2020 to spring 2021) of intensive care physicians in one of the two COVID-19 hub hospitals in Central Italy and aims to study the evolution of the mental health status of intensivists during the pandemic. Changes in their work activity due to the pandemic were studied anonymously together with their perception of organisational justice, occupational stress, sleep quality, anxiety, depression, burnout, job satisfaction, happiness, and intention to quit. In May–June 2021, one year after the baseline, doctors reported an increased workload, isolation at work and in their social life, a lack of time for physical activity and meditation, and compassion fatigue. Stress was inversely associated with the perception of justice in safety procedures and directly correlated with work isolation. Occupational stress was significantly associated with anxiety, depression, burnout, dissatisfaction, and their intention to quit. Procedural justice was significantly associated with happiness. Doctors believed vaccinations would help control the problem; however, this positive attitude had not yet resulted in improved mental health. Doctors reported high levels of distress (73%), sleep problems (28%), anxiety (25%), and depression (64%). Interventions to correct the situation are urgently needed.

## 1. Introduction

Worldwide, the physical and mental health of health care workers (HCWs) has been put at risk by the pandemic of coronavirus disease 2019 (COVID-19) caused by severe acute respiratory syndrome coronavirus 2 (SARS-CoV-2). During the first phase of the pandemic, HCWs who came into contact with patients and were not adequately protected developed the disease and in turn frequently became carriers of infection [1]. A systematic review of studies published before 8 July 2020 indicated that frontline HCWs frequently developed SARS-CoV-2 (the estimated cumulated prevalence of a positive reverse transcription-polymerase chain reaction on a mucosal swab was 11%, 95% confidence interval (CI): 7, 15) [2]. In that initial period, HCWs who were COVID-19-positive accounted for a significant proportion of all COVID-19 patients. Although the severity and mortality of the disease were lower among HCWs [3], several were affected by long COVID or had permanent outcomes.

In addition to experiencing physical consequences, HCWs also underwent dramatic psychological pressure that manifested itself in various ways during the different phases of the pandemic. An impressive number of scientific publications (to date, over two thousand studies and more than a hundred systematic reviews and meta-analyses) have helped us to understand what happened. Each of the many cross-sectional studies focused on a specific phase in the pandemic during which emerging problems were added to the usual stressors of medical activities, thus resulting in a high level of disorders such as post-traumatic stress, sleep problems, anxiety, depression, and burnout [4]. HCWs were exposed to a wide range of emotions and environmental conditions that varied over time. In the very early stages, HCWs were mainly concerned with defining strategies to treat an unknown disease and minimise the possibility of transmission (e.g., via air conditioning systems [5] inside hospitals) or finding new safety procedures to assist patients [6] as well as addressing the ethical dilemmas that emerged from the imbalance between care needs and resources during the COVID-19 pandemic [7] whereas in the subsequent recurring waves of the epidemic, the main stressors were prolonged periods of work in isolation, high workloads, compassion fatigue, and a lack of time for physical activity, meditation, or relaxation.

Clearly, a cross-sectional study is not able to report this complex series of varying emotional reactions that resulted in evident repercussions on the health of HCWs and consequently on the quality of care. A few research groups have carried out short prospective studies by repeatedly consulting HCWs anonymously to evaluate, for example, a possible reduction in stress levels between the initial phase of the pandemic and the following period [8,9] or adaptation to new safety measures [10,11]. Studies that had a longer duration witnessed a steep drop in participation: out of the thousands contacted, only a few dozen HCWs responded during the follow-up [12]. In the very extensive mental health literature concerning HCWs struggling with the pandemic, we have not been able to find long-term prospective studies that measure different aspects of mental health simultaneously.

This study, which began during the first phase of the pandemic, was designed to follow a group of workers who were continuously and exclusively engaged in the treatment of patients with COVID-19 in one of the two COVID-19 hub hospitals in Central Italy. Our aim was to measure the perception of organisational justice and occupational stress and how these varied in relation to external factors. To do this, we investigated their association with possible causal factors and the resulting consequences on sleep, anxiety, depression, burnout, happiness, job satisfaction, and the intention to quit.

## 2. Materials and Methods

### 2.1. Participants

All the anaesthetists working in the COVID-19 department of the “A. Gemelli” University hospital in Rome were invited to participate by completing an anonymous questionnaire on the SurveyMonkey online platform. The baseline collection was carried out in April–May 2020 during the first wave; a second collection was conducted in December 2020 during the second wave and the current collection was conducted in April–May 2021 during the third wave, exactly one year after the first. No incentives were provided for participation. The workers were informed by email of the results of the previous surveys and asked to participate. The survey was conducted in accordance with the Helsinki Declaration. The Catholic University Ethics Committee approved the study (ID 3292).

Of the 198 eligible workers who were in service on 1 April 2021, 120 completed the present survey (participation rate = 60.6%). The cohort varied because many workers who had participated in the baseline left the hospital in the course of the year. The percentage of trainees in the cohort increased significantly during our survey because the hospital hired them under fixed-term contracts, moving them from the general hospital where they served at the COVID-19 centre to meet the care needs posed by the pandemic. However, the age distribution of the cohort did not change during the survey. In the current survey, participants were mainly young (70% under 35 years of age), female (62, 51.7%) workers. Just over half of the participants (65, 54.2%) had been employed in the hospital for more than three years and 15 (12.5%) had been working there for less than a year.

The proportion of workers who reported unprotected exposure to COVID-19 patients increased significantly during the periods of observation (Table 1). At the time of the third survey, 59.2% reported at least one unprotected exposure. Of these, 4.2% occurred in a non-work environment and 16.9% both in the workplace and outside the workplace but, in most cases (78.9%), exposures were exclusively of a professional nature. A total of 23 HCWs (19.2%) had contracted COVID-19 and an additional 8 (6.7%) reported having had a false-positive antigen test at the periodic screening that all hospital workers undergo. A non-significant increase was observed in the prevalence of unprotected exposures and infections between the second and third survey. Most of the workers who had contracted the infection were completely asymptomatic (10, 38.5%) or had mild symptoms that did not require treatment (14, 53.8%); only 2 had mild symptoms that required home treatment. However, a significant proportion of the subjects who had contracted the disease reported protracted symptoms after the end of the infectious phase (long COVID, 38.5%) or permanent outcomes (post-COVID, 3.7%)

### 2.2. Questionnaire

The questionnaire used in the survey was composed of a series of ad hoc questions, mainly related to the phase of the pandemic, and a few standardised tools for measuring perceived organisational justice, stress, and effects on mental health. To facilitate the interpretation of the results, all the scales obtained from the questionnaire were standardised by dividing by the maximum value of the scale and multiplying by 100.

Organisational justice was measured with the Italian version [13] of the Colquitt questionnaire [14,15,16]. The workers were invited to assess the correctness of the safety procedures by means of 3-item questions (e.g., “Are these procedures error-free?”). Each question was answered according to a 5-point Likert scale, from 1 = “I totally disagree” to 5 = “I strongly agree”, thus producing a scale ranging from 3 to 15. In this study, the reliability of the questionnaire, measured by Cronbach’s alpha, was 0.749 (good). The raw score was standardised.

Stress was measured using the Italian version [17,18] of the Siegrist effort/reward imbalance model [19,20]. The questionnaire contained three graded questions on a 4-point Likert scale for effort and seven for the reward scale, thus constituting two scales graded from 3 to 12 and 7 to 28, respectively. The raw scores were standardised. The weighted ratio between effort and reward (effort reward imbalance, ERI), if greater than unity, indicated a state of distress. In this study, the reliability of the scales, measured by Cronbach’s alpha, was 0.726 for effort (good) and 0.820 for reward (very good).

Sleep quality was measured with the 2-item version of the “Sleep Condition Indicator” (SCI-02) [21,22], which aims to assess insomnia according to the Diagnostic Statistic Manual 5 (DSM5). Each question was graded on a 5-point Likert scale ranging from 4 to 0. The final score ranged between 0 and 8, with higher values indicating a better sleep quality. A score of ≤4 revealed a possible insomnia disorder. Cronbach’s alpha in this study was 0.746 (good). The raw scores were standardised.

Mental health was measured using Goldberg’s anxiety and depression scales (GADS) [23,24], each of which consisted of 9 binary questions. A score of 5 or more affirmative answers to the questions on the anxiety scale and two or more to the questions on the depression scale indicated a probable diagnosis of anxiety and depression. In this study, the reliability of the GADS questionnaire, measured by Cronbach’s alpha, was 0.788 (good).

Job satisfaction was measured by a single question expressed on a 7-point Likert scale ranging from extremely dissatisfied to extremely satisfied, according to Warr et al. [25,26]. Happiness was measured by the 10-point scale of Ab-del-Khalek [27]. The frequency of burnout feelings was measured on a 6-point scale, according to West et al. [28]. The intention to quit the hospital was measured with a single item (yes/no).

### 2.3. Statistics

The variables were analysed in descriptive terms of mean and standard deviation for continuous variables and frequency for categorical variables. The variables measured at the baseline during the first pandemic wave (T_0_), the second wave (T_1_), and the third wave (T_2_) were compared by an analysis of variance and a post-hoc comparison using the Bonferroni test if continuous or by means of the chi-squared and Fischer test if categorical.

A stepwise linear regression was used to determine which of the possible stressors had a greater effect on occupational stress. Perceived stress was the dependent variable (effort-reward imbalance). The independent variables included in the model were gender, age range, physical activity, meditation, procedural justice, workload, monotony, compassion fatigue, isolation at work, and social loneliness. In the stepwise selection method, the model started by entering the variable with the smallest *p*-value (PIN *p* < 0.05); after each step in which a variable was added, all candidate variables in the model were checked to see if their significance had fallen below the specified tolerance level (POUT *p* > 0.1).

To study the association of perceived justice and stress with mental health indicators, we constructed multiple linear regression models to assess the effect on sleep quality, anxiety, and depression and adjusted the result for age and gender.

Finally, we assessed the extent to which working conditions determined possible cases of anxiety, depression, burnout, dissatisfaction, and the intention to leave the workplace by constructing multiple logistic regression models adjusted for gender and age. In this way, we calculated the odds ratio (OR). For each OR, we calculated the 95% confidence interval (CI95%).

The analyses were performed using IBM/SPSS 26.0 (IBM Corporation, Armonk, NY, USA).

## 3. Results

In the third survey, the subjective perception of workload tended to grow progressively. The workers confirmed that their workload was greater/much greater than before the pandemic. For many of them, the type of medical activity had also become progressively more repetitive and monotonous because of the need to continually apply the same diagnostic and therapeutic procedures in COVID-19 patients. For safety reasons, contact with their patients’ families was limited and there was an increasingly frequent need to inform patients of the unfavourable outcome of treatment, all of which contributed to determining compassion fatigue (Table 2). All these unfavourable occupational changes were reported more frequently in this survey than at the baseline. Moreover, 40% of workers complained of having to work in isolation and about 70% suffered from a reduction in social contacts. However, between the second and third surveys, we observed a significantly lower frequency of workers who complained of isolation in their social life. Factors that contribute to increasing resilience such as the time devoted to physical activity, meditation, prayer, or spiritual activities were reduced or greatly reduced in most workers, as in previous surveys (Table 2).

All workers were vaccinated between the second and third surveys. Most of them were moderately or strongly in agreement (71.0%) with the following statement: “With vaccinations it will be possible to control the pandemic”.

The perception of procedural justice, i.e., the degree of trust in safety measures, was not high, exactly as in the previous surveys (Table 3).

On average, the efforts made by workers to respond to job demands remained very high (77% of the maximum value), confirming the level recorded in the second survey, which was significantly higher than at the baseline. The rewards earned from work showed a modest, non-significant increase. Occupational stress levels were on average much higher than the equivalence level between efforts and rewards, indicating a widespread state of distress in the sample. The share of distressed workers remained constant in the three surveys: at least three out of four workers were in a state of distress throughout the year.

The average score of the GADS anxiety scale did not register significant changes in the third survey and therefore it was confirmed that more than one in four workers had a score corresponding with a diagnosis of anxiety made by a specialist. Conversely, the mean score of the depression scale showed a significant increase in the second survey compared with the baseline; in the present survey, it remained constant. Three out of five workers manifested depressive symptoms.

The quality of sleep, although remaining rather low (scores on average at two thirds of the maximum) showed a slight, non-significant improvement in the third survey compared with the baseline. The number of workers affected by insomnia was significantly lower in this survey than during the first wave.

A stepwise linear regression analysis was conducted to evaluate which of the variations in work activity associated with COVID-19 was most closely related to occupational stress. The prediction model, which explained 39.4% of the variance of stress, included isolation at work and a reduced perception of organisational justice in addition to the age group > 35 years (Table 4).

The perception of organisational justice and the occupational stress variables were significantly associated with poor sleep quality, anxiety, and depression. In particular, the effort made to work was significantly associated with a reduced quality of sleep and with an increased anxiety and depression score in a multiple linear regression model adjusted for demographic variables (Table 5).

A total of 21% of workers said they were dissatisfied with their job and 41.2% said they intended to quit. The average happiness score was 6.55 ± 1.92 on a scale of 1 to 10. Nearly half the workers (46.5%) reported experiencing burnout several times a month or more frequently.

The relationship between stress and the perception of justice and mental health was studied using a logistic regression analysis. The risk of being anxious and depressed or suffering from burnout was significantly associated with effort whereas the intangible rewards derived from work (reward) were protective towards burnout, job dissatisfaction and the intention to quit. Dissatisfaction with one’s job and the intention to leave the job were significantly associated with high effort and low reward. Happiness was significantly associated with organisational justice (Table 6).

## 4. Discussion

This study, which, to the best of our knowledge, is the only prospective research on intensive care HCWs caring exclusively for COVID-19 cases conducted over a period of one year starting from the beginning of the pandemic, has shown that the mental health status of these workers is not excellent. Occupational stress, which remained high throughout the observation period, was associated with an elevated frequency of anxiety and an increasing prevalence of depression. Nearly half of the workers often felt burnout, and levels of job satisfaction and happiness in life were not satisfactory. A considerable number of intensivists planned to leave the hospital.

During epidemics, frontline anaesthetists are among the most vulnerable HCWs on account of infections and mental health problems [29,30,31,32]. All the effects observed in our sample have been reported by other cross-sectional studies on HCWs engaged on the frontline during the pandemic. Insomnia, anxiety, and depression were observed in the early phases of the pandemic in Chinese workers [33]. Fear and lack of confidence in safety measures were associated with reduced job satisfaction and the intention to leave the job [34]. Later on, these negative emotions occasionally led to post-traumatic stress disorder [35,36] or burnout [37,38]. The psychological picture naturally varied over time; a few months after the acute phase of the epidemic, both recurring involuntary memories and happiness were described [39]. The type of occupational problems to which workers were exposed changed over the course of the pandemic: in the early stages, a lack of readiness, a shortage of PPE, separation from families, stigma [40], and an increased workload [41,42] prevailed among professionals whereas in later stages other stressors, such as the death of patients and colleagues inducing moral injury and distress [43,44] and isolation or lack of support at work [45,46], attracted the attention of researchers. Moreover, a lack of physical activity has been associated with a poor quality of life in frontline HCWs [47], and a number of studies has underlined the importance of meditation and spirituality in improving psychological resilience in HCWs during the pandemic [48,49,50,51].

The repeated cross-sectional nature of our epidemiological design enabled us to follow variations over time in the response of HCWs to the pressure posed by the pandemic. Our setting—one of the two hub hospitals for COVID-19 in Central Italy—was typical of the conditions observed throughout the country. During the first wave of the pandemic in Italy in the spring of 2020 [52], the shortage of personal protective equipment (PPE), the fear of infection, and uncertainty about new safety measures were the main stressors [53] especially for younger workers and residents [54]. Before widespread screening measures were introduced [55], the oligosymptomatic carriers of SARS-CoV-2 represented a particularly threatening occupational risk that was difficult to predict [56]. HCWs who experienced unprotected exposure to patients with Covid-19 and, to a greater extent, those who tested positive for PCR manifested elevated levels of anxiety, depression, and sleep disturbances [57]. Our prospective observation of a highly selected sample of workers continuously engaged in caring for COVID-19 patients demonstrated that in the first phase of the pandemic, the main stressors were the need to adhere to new safety procedures and uncertainty about their effectiveness [58]. The younger and less experienced residents complained of a significantly lower level of informational justice than the specialists although they had all undergone the same training [54]. Confidence in the correctness of safety procedures immediately proved to be an important factor in protecting against occupational stress. The widespread state of alarm and fear for their own health and that of their family members strongly influenced the quality of sleep [58].

During the second wave, in the autumn of 2020 [59], when the question of protective devices had been solved and new safety procedures were in place, other problems became evident. Difficulties in relations between doctors and patients’ relatives led to a sharp deterioration in public opinion towards doctors, as witnessed by a surge in complaints of malpractice [60]. The availability of effective and rapid screening tests made possible a better control of infections; however, this continued to affect HCWs and thus reduced the workforce even in sectors where the workload was already excessive. The isolation of patients from their relatives and the isolation of HCWs from their colleagues proved to be a major stressor. Frontline HCWs were strongly isolated in their social life and registered a strong change in the orientation of public opinion towards them, which passed in a few months from very favourable to critical [60]. In our sample, the high workload and lack of time for meditation and activities that allow for mental recovery have been, in the opinion of doctors, increasingly important stressors. Their work was always carried out in solitude. The relationships with the patients’ relatives became less but paradoxically the need to inform them of the unfavourable outcome of the therapies increased. This has certainly contributed to changing opinion towards doctors and has increased their social isolation. The prolongation of the epidemic—with workload levels that were higher than at the baseline without time to devote to family, sports, or meditation and persistent uncertainty about the correctness and effectiveness of safety procedure—has led to a significant increase in symptoms of depression [61].

In early 2021, the availability of vaccines made it possible to vaccinate all HCWs who consequently perceived the possibility of controlling the pandemic. Immunised workers probably felt able to resume social activities. In fact, the third survey reported a reduced prevalence of those who complained of isolation in life. However, at the time of our investigation, these positive changes had still not had a significant impact on mental health conditions. Only sleep quality showed a modest improvement from the baseline whereas distress, anxiety, and depression remained unacceptably high. Nevertheless, the trend towards improved sleep is worth highlighting because sleep has been shown to be a moderator of the relationship between stress and mental health [57] and could therefore be a positive indication of possible future health improvements. The factors that weigh most heavily on the perception of stress at this moment are isolation at work and the perception of a lack of correctness in the organisation of work. A year after the outbreak of the coronavirus epidemic, older workers such as specialists with permanent contracts are shouldering the greatest burden, probably because during the current stable phase of the epidemic, they are responsible for organisation and training.

Clearly, the situation illustrated in our study calls for preventive and supportive action. Unfortunately, it is far from easy to implement this kind of intervention. Excessive workload could be remedied by increasing staff but adequately trained personnel are not available and, as we have seen, the hiring of young physicians leads to training problems [54]. Preventive social distancing hinders clinical training and relationships with patients’ relatives, thus increasing the clinical risk and the danger of a reduction in the quality of care. The lack of time to devote to physical activity or meditation and intellectual activities reduces resilience and hinders the application of individual psychological support treatments. The high percentage of workers reporting unprotected exposures and the fact that one in five has contracted COVID-19 indicate the need to improve safety procedures and their enforcement. The pandemic has compelled hospital authorities to introduce safety measures with a “top-down” approach. The low degree of confidence in these procedures, which still persists a year after their implementation, should encourage the authorities to obtain greater worker participation in the planning and control of these measures. A “bottom-up” approach involving participatory ergonomics groups [62] could increase the collaboration of workers, the effectiveness of the measures, and the perception of organisational justice, thus reducing occupational stress. Another administrative measure that could reduce the perception of stress (if not effort) would be to increase material and immaterial rewards that doctors receive for their work. Furthermore, given the importance of sleep in the relationship between stress and pathologies [63,64], the utmost attention should be given to scheduling work shifts and respecting recovery times. Workers should be informed about the importance of proper sleep hygiene and trained to prevent sleep disturbances. This simple measure has proved effective in preventing stress damage in other categories of workers [65].

This study has several limitations. Although it was conducted over a one-year period on a high-risk population simultaneously investigating numerous variables that make up the complex relationships between work, stress, and health, our study was limited by being able to observe only one setting and therefore a numerically modest sample. The chosen setting, one of the two COVID-19 hospitals in Central Italy, and the representativeness of the response, which in each phase of the longitudinal study involved a qualified majority of those eligible, authorise us to believe that the results realistically describe the situation of workers continuously and exclusively dedicated to treating COVID-19 patients from the beginning of the pandemic to today. However, we cannot assume that the findings can be applied to all HCWs.

Another limitation is related to the epidemiological model. As participants were guaranteed anonymity, we were unable to evaluate the incidence of the reported pathologies; however, the prospective nature of the observations, which were repeated in correspondence with the three pandemic waves, made it possible to describe the evolution of the psychological state of frontline physicians during COVID-19 with greater effectiveness than in the numerous cross-sectional investigations conducted around the world.

## 5. Conclusions

In conclusion, our study documented the complexity and relevance of the psychological response of physicians at the forefront of the COVID-19 pandemic. Workers responded to the uncertainties and unpreparedness of the first wave with anxiety and sleep problems. The protracted work in isolation, the lack of time for meditation, and growing compassion fatigue resulted in a significant increase in depression in the second phase. In the third phase, the availability of vaccines allowed a partial resumption of social contacts but workers still reported concerns about safety measures, excessive workload, responsibility, high occupational stress, anxiety and depression, low satisfaction, burnout, and the intention to quit. The picture that emerged from one year of observations calls for the adoption of support measures. Participatory involvement in safety procedures, increased intangible rewards, and increased attention to meditation and sleep are recommended.

If the photo symbolising healthcare in Italy in the spring of 2020 was that of a nurse falling asleep in the workplace [66]—thus illustrating both the self-denial of the individual and the inadequacy of the work organisation—today, it is fair to ask that doctors who provide intensive care for COVID-19 patients have full occupational well-being.

## Figures and Tables

**Table 1 ijerph-18-09888-t001:** Characteristics of the population.

Variables	Baseline		2nd Survey		3rd Survey		X^2^
	*n*	%	*n*	%	*n*	%	*p*
Participant	154		105		120		
Resident	58	37.7	55	52.4	68	56.7	0.004
Gender, male	75	48.7	51	48.6	58	48.3	0.998
Age, < 35 years	94	61.0	76	72.4	84	70.0	0.115
Reporting unprotected exposure to COVID-19 patients	38	24.7	59	56.2	71	59.2	0.000
Reporting a false-positive swab test	-	-	2	1.9	8	6.7	-
Reporting COVID-19 disease	-	-	16	15.2	23	19.2	0.437
Asymptomatic COVID-19 case	-	-	6	37.5	10	38.5	0.709
Mild COVID-19 case	-	-	9	56.3	14	53.8	0.773
Moderate COVID-19 case	-	-	1	6.3	2	7.7	-
Reporting long COVID	-	-	-	-	10	38.5	-
Reporting post-COVID	-	-	-	-	1	3.7	-

**Table 2 ijerph-18-09888-t002:** Changes reported during the COVID-19 outbreak and prevalence of high stress, insomnia, anxiety, and depression during the 1st and 2nd waves.

Reported Effect	Baseline		2nd Survey		3rd Survey		
	*n*	%	*n*	%	*n*	%	*p*
Increased/greatly increased workload	77	50.0	83	83.0	98	84.5	0.000
The work became more repetitive and monotonous	51	33.1	36	36.0	53	45.7	0.162
More frequent need to inform of the death of a relative	61	39.6	65	65.0	81	69.8	0.000
Isolation at work			42	42.0	47	40.5	0.669
Isolation in life			81	81.0	78	67.2	0.008
Time for physical exercise was shorter/much shorter	117	76.0	80	80.0	92	79.3	0.742
Time for meditation was shorter/much shorter	72	46.8	65	65.0	74	63.8	0.006
Distressed (effort/reward weighted ratio > 1)	117	76.0	80	80.0	83	72.8	0.468
Insomniac (SCI08 score ≤ 16; SCI02 score ≤ 4)	58	43.3	33	33.0	32	28.1	0.037
Anxious (GADS anxiety score ≥ 5)	40	26.0	31	31.0	29	25.4	0.599
Depressed (GADS depression score ≥ 2)	75	48.7	63	63.0	73	64.0	0.017

SCI08 = Sleep Condition Indicator used in the baseline survey; SCI02 = Sleep Condition Indicator short form with two items used in the 2nd and 3rd survey; GADS = Goldberg’s anxiety and depression scales.

**Table 3 ijerph-18-09888-t003:** Mental health indicators (perceived justice, occupational stress, sleep quality, anxiety, depression) in anaesthesiologists during the three waves of the COVID-19 outbreak.

Variable	1st Wave	2nd Wave	3rd Wave	ANOVA	Bonferroni	
	Mean ± s.d.	Mean ± s.d.	Mean ± s.d.	*p*	*p*	
Procedural Justice	49.91 ± 13.64	53.60 ± 15.60	53.33 ± 15.67	0.079		
Effort	71.48 ± 16.59	77.91 ± 14.03	77.34 ± 14.52	0.001	1 vs. 2 0.003	1 vs. 3 0.006
Reward	58.88 ± 13.13	59.36 ± 13.95	61.40 ± 13.97	0.304		
Job stress	1.30 ± 0.51	1.42 ± 0.56	1.37 ± 0.57	0.228		
Sleep quality	59.64 ± 25.11	65.13 ± 28.50	67.43 ± 27.31	0.051		
Anxiety	3.04 ± 2.32	3.34 ± 2.33	3.02 ± 1.93	0.487		
Depression	1.97 ± 1.87	2.71 ± 1.95	2.49 ± 1.91	0.007	1 vs. 2 0.008	1 vs. 3 n.s.

**Table 4 ijerph-18-09888-t004:** Third wave stepwise linear regression analysis: the relationship between job changes and perceived work-related stress (ERI).

Variable	ERI	
	Standardised Beta	*p*
Isolation at work	0.383	0.000
Procedural justice	−0.335	0.000
Age class.	0.293	0.000
Determination coefficient of the model (R^2^)		0.394

Variables excluded from the model: gender, monotony, compassion fatigue, social loneliness, physical activity, workload, and meditation.

**Table 5 ijerph-18-09888-t005:** Third wave health outcomes associated with procedural justice and occupational stress: a linear regression analysis adjusted for age and gender.

Variable	Sleep Quality		Anxiety		Depression	
	Standardised Beta	*p*	Standardised Beta	*p*	Standardised Beta	*p*
Procedural justice	0.062	0.628	−0.022	0.845	0.027	0.810
Effort	−0.333	0.013	0.541	0.000	0.578	0.000
Reward	0.066	0.613	−0.084	0.473	−0.057	0.622

**Table 6 ijerph-18-09888-t006:** Third wave health outcomes associated with procedural justice and occupational stress: a multivariate logistic regression model adjusted for age and gender.

Predictor	Dependent Variable OR (CI95%)					
	Anxious ^1^	Depressed ^2^	Burned Out ^3^	Dissatisfied ^4^	Happy ^3^	Intention to Quit
Procedural justice	1.063 (0.843–1.340)	0.905 (0.729–1.124)	1.062 (0.840–1.344)	0.970 (0.765–1.230)	1.252 * (1.000–1.568)	0.870 (0.675–1.121)
Effort	1.721 (1.199–2.468) **	1.515 (1.095–2.096) *	2.151 (1.435–3.224) ***	1.459 (1.027–2.071) *	0.765 (0.560–1.044)	1.871 (1.230–2.847) **
Reward	0.947 (0.825–1.087)	0.926 (0.812–1.057)	0.848 (0.737–0.975) *	0.766 (0.655–0.894) ***	1.049 (0.916–1.201)	0.762 (0.650–0.892) ***

^1^ = GADS anxiety score ≥ 5; ^2^ = GADS depression score ≥ 2; ^3^ = dichotomised at the median; ^4^ = moderately, very, or extremely dissatisfied. * *p* < 0.05; ** *p* < 0.01; *** *p* < 0.001.

## Data Availability

Data deposited on Zenodo: DOI Number 10.5281/zenodo.5516265. The study protocol will be provided upon request.

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
