# Peer review of "A One-Year Prospective Study of Work-Related Mental Health in the Intensivists of a COVID-19 Hub Hospital"

_ijerph, 2021, doi:10.3390/ijerph18189888_

Round 1
Reviewer 1 Report
This is a sound paper that painstakingly follows the ideal format and prescriptions of quantitative research. One of the traps in this type of medical research is the danger of unintentionally and unobtrusively disregarding minor ethical procedural rules. The danger looms larger and the pitfalls deeper in a research that is spread out over a period of one year. This research does not fall into that trap and it avoided areas of danger. This is because the authors avoided taking shortcuts. They pursued the research in an ideal and rigorous manner that I normally teach in my graduate research course at the university. The paper is both scientifically and ethically sound. I also like the honesty in expressing gratitude to E.A. Wright "who revised the English language" (p. 10). He did an excellent job. Any little error like "COVID19" instead of "COVID-19" (p. 9) can be readily corrected by the editor.
However, I wish you had included in your introduction the dilemma that Italian doctors and healthcare workers faced, and the worldwide attention Italy drew during the first wave of COVID-19 when Luigi Riccioni, the anesthesiologist who was the head of the Italian Society of Anesthesia, Analgesia, Resuscitation and Intensive Care gave guidelines to doctors to prioritize treatment for COVID-19 patients that allegedly favored younger people over elderly ones. Giving that as part of the introduction would heighten the difficult ethical choice doctors and healthcare workers faced. That then helps to highlight the mental and physical toll on healthcare workers that would manifest itself in the conditions examined in the paper. Whether you decide to add that bit of information or not, the paper remains excellent and strong.
Author Response
This is a sound paper that painstakingly follows the ideal format and prescriptions of quantitative research. One of the traps in this type of medical research is the danger of unintentionally and unobtrusively disregarding minor ethical procedural rules. The danger looms larger and the pitfalls deeper in a research that is spread out over a period of one year. This research does not fall into that trap and it avoided areas of danger. This is because the authors avoided taking shortcuts. They pursued the research in an ideal and rigorous manner that I normally teach in my graduate research course at the university. The paper is both scientifically and ethically sound. I also like the honesty in expressing gratitude to E.A. Wright "who revised the English language" (p. 10). He did an excellent job. Any little error like "COVID19" instead of "COVID-19" (p. 9) can be readily corrected by the editor.
Response: We thank the reviewer for the words that encourage us to continue our work. We have corrected the reported error
However, I wish you had included in your introduction the dilemma that Italian doctors and healthcare workers faced, and the worldwide attention Italy drew during the first wave of COVID-19 when Luigi Riccioni, the anesthesiologist who was the head of the Italian Society of Anesthesia, Analgesia, Resuscitation and Intensive Care gave guidelines to doctors to prioritize treatment for COVID-19 patients that allegedly favored younger people over elderly ones. Giving that as part of the introduction would heighten the difficult ethical choice doctors and healthcare workers faced. That then helps to highlight the mental and physical toll on healthcare workers that would manifest itself in the conditions examined in the paper. Whether you decide to add that bit of information or not, the paper remains excellent and strong.
R.: The argument that the reviewer points out is of extreme importance and certainly deserves at least a mention. We have included the reference to this type of problem in the introduction.
Reviewer 2 Report
It is worth paying attention to the effects that the Covid19 pandemic has had on health care workers, especially those who have been on the front lines. In this sense, the work presented is of enormous relevance and interest. Moreover, it does not only stay in a single moment but is able to trace how the evolution in some aspects related to occupational health has occurred in a retrospective way.
Below we present a series of questions that could help to improve the work presented:
- It would be interesting if the title of the manuscript were better adjusted to the objectives of the research conducted.
- The tools do not mention how the intention to quit smoking is measured.
- The authors incorporate a significant amount of data on participants that could be better placed in the results.
- The authors should specify how ethical aspects were considered in the work performed.
- The conclusions are linked to the discussion and should appear in a different section.
Author Response
It is worth paying attention to the effects that the Covid19 pandemic has had on health care workers, especially those who have been on the front lines. In this sense, the work presented is of enormous relevance and interest. Moreover, it does not only stay in a single moment but is able to trace how the evolution in some aspects related to occupational health has occurred in a retrospective way.
R.: We thank the reviewer for the appreciation expressed for our work and for the time he/she devoted to the review
Below we present a series of questions that could help to improve the work presented:
- It would be interesting if the title of the manuscript were better adjusted to the objectives of the research conducted.
R.: We gladly accepted the reviewer's suggestion. As also pointed out by reviewer #3, we agree that the term "occupational health" is too vague. We, therefore, adopted the term “work-related mental health” and changed the title into: “One-year prospective study of work-related mental health in the intensivists of a COVID-19 hub hospital”
- The tools do not mention how the intention to quit smoking is measured.
R.: In this specific work we have not evaluated the use of tobacco. We assessed the intention to leave the workplace with a single question with a binary answer (yes / no), as specified in the last line of paragraph 2.2
- The authors incorporate a significant amount of data on participants that could be better placed in the results.
R.: The author is right, many data on the description of the population could be considered as results of a survey and not as starting data. For example, the fact of having had unprotected exposure to COVID-19 patients, or having contracted the infection, and the course of this infection, would certainly be the results of a clinical investigation. However, this study focuses on workers' mental health. For this reason, in the baseline report on the same study, a reviewer advised us to remove this information from the results and place them in the description of the case history. In this study, we preferred to keep this approach, both for homogeneity with previous works, and because we are convinced that in this way readers will be led to focus mainly on mental health.
- The authors should specify how ethical aspects were considered in the work performed.
R.: The ethical authorization procedure for studies conducted in our university involves an initial evaluation by the Department Council, then an evaluation by the Scientific Direction of the Polyclinic and finally an evaluation by the Ethics Committee. The required documentation includes, in addition to a detailed research project, the informed consent form, and various other documents that vary according to the type of study. At the beginning of this research, the doctors working in the emergency department of the hub hospital were personally contacted by one of the authors, who informed them of the research and submitted the informed consent form to everyone. The doctors then received an email indicating the link to the data collection program (SurveyMonkey). The questions were constructed in such a way that it was not possible to know who answered. We have indicated in the article, as in the previous ones, the authorization number of the Ethics Committee. We have provided the Journal with a copy of the consent form. The detailed research project will be provided to readers who request it from the authors. We have added this information in the Data Availability Statement section.
- The conclusions are linked to the discussion and should appear in a different section.
R.: We agree with the reviewer's suggestion. We have placed the conclusions in a different paragraph. We are grateful for this observation which allowed us to better write the conclusions.
Reviewer 3 Report
Thank you for the opportunity to read this very interesting paper. The paper has many strengths that should be of interest to the journal audience. Thus, the following suggestions are around enhancing the presentation for publication and clarifying aspects of the data and reporting.
I will go by line number for the most part. If not, I will try to be as specific as possible in noting the area I am speaking about.
Abstract
It is necessary to include the main aim.
Methods. Which was the sample? Which were the instruments used?
Results. According the sentence “Occupational stress was significantly associated with anxiety, depression, burnout, …”. I think that authors must include p-values
Keywords. COVID-19
- Introduction
Authors must speak more about the adverse working conditions affecting the mental health, which are the consequences of high levels of psychological distress (professionals and patients), etc. Are there in these environments a high presence of symptomatology related to work stress (physical and emotional fatigue, overload, tension, and anxiety) that may pose a risk of impaired mental health? Why? Are the differences between anesthetists VS other health care workers?
Authors must finish the Introduction Section with the main aim.
- Materials and Methods
Did the authors calculate the needed sample size? Please, clarify. How was the sample size determined? Did the authors test power calculation?
How was the sample chosen? Which is the total population? Authors must specify it.
Do the authors have a study protocol? The study protocol should be described in detail.
- Discussion
Limitations related with the type of methodology used. Limitations regarding representativeness of respondents should be better addressed Authors must specify it. The fact of having a convenience sample should be included in the limitations of the study.
Conclusions are not related with the main aim
I wish you all the best.
Author Response
Thank you for the opportunity to read this very interesting paper. The paper has many strengths that should be of interest to the journal audience. Thus, the following suggestions are around enhancing the presentation for publication and clarifying aspects of the data and reporting. I will go by line number for the most part. If not, I will try to be as specific as possible in noting the area I am speaking about.
R.: We thank the reviewer for the appreciation of our work. The advice given was very helpful to us.
Abstract. It is necessary to include the main aim.
R.: The observation is timely. We added the sentence "and aims to study the evolution of the mental health status of intensivists during the pandemic".
Methods. Which was the sample? Which were the instruments used?
R.: The sample was described in Chapter 2.1, Participants. There were 198 eligible workers (Line 94). The survey tool was described in Chapter 2.2, Questionnaire; it was composed of some ad hoc questions and some standardized questionnaires.
Results. According the sentence “Occupational stress was significantly associated with anxiety, depression, burnout, …”. I think that authors must include p-values
Response. The reviewer is right, and we would certainly have gladly satisfied him/her if the statistical tests were few. The associations between perceived justice and stress and the different outcomes are shown in tables 5 and 6, which contain 9 and 18 p values, respectively. We considered the indication of the tables sufficient because reporting all the values in the text would have made it too heavy and difficult to understand.
Keywords. COVID-19
Response: We agree that COVID-19 is the main topic and in fact, we mention this term in the title. In accordance with the editorial rules, we have not reported in the keywords the same terms that appear in the title
Introduction. Authors must speak more about the adverse working conditions affecting mental health, which are the consequences of high levels of psychological distress (professionals and patients), etc. Are there in these environments a high presence of symptomatology related to work stress (physical and emotional fatigue, overload, tension, and anxiety) that may pose a risk of impaired mental health? Why? Are the differences between anaesthetists VS other health care workers?
R.: With this acute observation, the reviewer actually anticipated an idea that we had to extend the study in the future to other groups of workers in the same hospital, to verify if they too have a high presence of symptomatology related to work stress like the occupational situation we have observed in this COVID-19 ICU study.
Authors must finish the Introduction Section with the main aim.
R.: According to what the reviewer has indicated, we have concluded the introduction by indicating, in the last four lines, the purposes of the study.
Materials and Methods. Did the authors calculate the needed sample size? Please, clarify. How was the sample size determined? Did the authors test power calculation?
R.: Since the study is descriptive and aims to measure the prevalence of anxiety, depression and stress, the sample size is not critical. The most critical aspect concerns the participation in the study which, if too low, would detract from the meaning of the data collection. In this study, we have always obtained a participation rate of over 60% of those eligible to participate. However, with a population of 198 members, and a confidence level of 95%, when we choose a confidence interval of 6 we need 114 responses, which is below the number of responses obtained [https://www.surveysystem.com/sscalc.htm]. Given the dimension of the population and the participation rate, the confidence interval was 5.63.
How was the sample chosen? Which is the total population? Authors must specify it.
R.: The sample is made up of the entire population of intensivists serving in one of the two COVID-19 hub centres in Central Italy. The number of intensivists on duty during the third wave, 198, is reported in the article on line 94, the participation rate, 60.6%, on line 95
Do the authors have a study protocol? The study protocol should be described in detail.
R.: The prospective study is composed of a series of online surveys conducted in correspondence with the first pandemic wave (April-May 2020), the second (December 2020) and the third (May 2021). The method is repeated cross-sectional. The population and tool used are detailed in the material and methods section. Further information on the first two phases of the investigation can be found in the articles published in this journal and cited here. The study protocol will be provided upon request. We have added this information in the Data Availability Statement section.
Discussion. Limitations related with the type of methodology used. Limitations regarding representativeness of respondents should be better addressed Authors must specify it. The fact of having a convenience sample should be included in the limitations of the study.
R.: In the Discussion, we have indicated these limitations. The first is the setting. We have added, as suggested by the reviewer, a series of considerations on this element. The text is now as follows:
“This study has several limitations. Although it was conducted over a one-year period on a high-risk population, simultaneously investigating numerous variables that make up the complex relationships between work, stress and health, our study was limited by being able to observe only one setting and therefore a numerically modest sample. The chosen setting, one of the two COVID-19 hospitals in Central Italy, and the representativeness of the response, which in each phase of the longitudinal study involved a qualified majority of those eligible, authorize us to believe that the results collected realistically describe the situation of workers continuously and exclusively dedicated to treating COVID-19 patients from the beginning of the pandemic to today. However, we cannot assume that the findings can be applied to all HCWs. Another limitation is related to the epidemiological model. Because participants were guaranteed anonymity, we were unable to evaluate the incidence of the reported pathologies; however, the prospective nature of the observations, which were repeated in correspondence with the three pandemic waves, made it possible to describe the evolution of the psychological state of frontline physicians during COVID-19 with greater effectiveness than in the numerous cross-sectional investigations conducted around the world.”
Conclusions are not related with the main aim. I wish you all the best.
R.: We are grateful for this observation which allowed us to better write the conclusions. Also following the advice of reviewer # 2, we have separated the paragraph of the Conclusions from the previous one and we have given it a character more in line with the topics covered in the article. The text now is as it follows: “5. Conclusions
In conclusion, our study documented the complexity and relevance of the psychological response of physicians at the forefront of the COVID-19 pandemic. Workers responded to the uncertainties and unpreparedness of the first wave with anxiety and sleep problems. The protracted work in isolation, the lack of time for meditation, the growing com-passion fatigue resulted in a significant increase in depression in the second phase. In the third phase, the availability of vaccines has allowed a partial resumption of social contacts, but workers still reported concerns about safety measures, excessive workload and responsibility, high occupational stress, anxiety and depression, low satisfaction, burnout and intention to quit. The picture that emerged from one year of observations calls for the adoption of support measures. Participatory involvement in safety procedures, increased intangible rewards, increased attention to meditation and sleep are recommended.
If the photo symbolizing healthcare in Italy in the spring of 2020 was that of a nurse falling asleep in the workplace [61], thus illustrating both the self-denial of the individual and the inadequacy of work organization, today it is fair to ask that doctors who provide intensive care for COVID-19 patients have full occupational well-being.”
Reviewer 4 Report
This is an interesting and up to date study on occupational health in times of COVID-19. Nevertheless, there are a number of issues that require further clarification which we will list below:
- The study is presented as being concerned with the occupational health of health care workers (see title). However, the concept of occupational health is never addressed throughout the paper, nor is it even included in the group of key words. In addition, one of the concepts adopted was "health care workers"; however, the sample is composed exclusively of anaesthesiologists. Being healthcare workers, they are a particular group and if the study is about them, this should be explicit in the paper. Curiously, in the chapter regarding the discussion of data, the comparison is made with studies carried out with representative samples of the various health professions.
- In the chapter on "Participants", it is mentioned that the sample was composed of 120 anaesthesiologists. However, as the authors state "The cohort varied since many workers who had participated in the baseline left the hospital in the course of the year". The authors add "The percentage of trainees in the cohort increased significantly during our survey because the hospital hired them under fixed-term contracts to meet the care needs posed by the pandemic". Despite this, it is not explicitly stated which number of hired trainees integrated the sample. This is of total relevance considering the nature of the concepts and variables under study. Let's see: assuming that hired trainees were not previously part of the hospital, they will need an adequate work environment to develop their activity. If they do not find such an environment, particularly in a situation such as the one underlying the various waves of the pandemic, they will be particularly exposed both from a professional and personal point of view. Therefore, it is essential to understand the relative weight of hired trainees in the global sample, and if there are differences in the results according to the length of service and the fact of being a hired trainee. This is particularly relevant because, as the authors state "The excessive workload could be remedied by increasing staff, but adequately trained personnel are not available and, as we have seen, the hiring of young physicians leads to training problems".
- Finally, table 2 shows some results that need some additional explanation. Let us see: at the same time that the workload increased and the work became more repetitive and monotonous, isolation at work decreased slightly and isolation in life decreased significantly; however, the Factors that can contribute to increasing resilience such as the time devoted to physical activity, meditation, prayer or spiritual activities were reduced or greatly reduced in most workers; conclusion: there seems to be some inconsistency in these data or else there are other variables that will help to better understand them.
Author Response
This is an interesting and up to date study on occupational health in times of COVID-19. Nevertheless, there are a number of issues that require further clarification which we will list below:
Response: We thank the reviewer for paying attention to our work. Her/his observations stimulated us to deepen and clarify some aspects of the study.
The study is presented as being concerned with the occupational health of health care workers (see title). However, the concept of occupational health is never addressed throughout the paper, nor is it even included in the group of keywords. In addition, one of the concepts adopted was "health care workers"; however, the sample is composed exclusively of anaesthesiologists. Being healthcare workers, they are a particular group and if the study is about them, this should be explicit in the paper. Curiously, in the chapter regarding the discussion of data, the comparison is made with studies carried out with representative samples of the various health professions.
Response: The reviewer is right. We changed the title, preferring the term "work-related mental health" to that of occupational health. Having removed the term "occupational health" from the title, we have placed it in the keywords. In fact, we believe that this is an occupational medicine study, both because the mental health of workers is part of the broader concept of occupational health, and because in investigating the effects of the pandemic we have also taken into consideration the infectious and clinical aspects. In the article, we take into account the share of workers who have contracted COVID-19, the clinical course of the infection and the outcomes. Since the focus of this research was mental health, we incorporated these data into the population description, thus considering them a pre-condition to mental health itself. In our previous study, we have in fact observed how the infectious state is associated with mental well-being [see ref. 53, Magnavita et al.].
The reviewer's remark about the distinction between intensivists and other health workers is precise. Where necessary, we specified that our sample consisted of intensive care HCWs. In the literature, there are numerous studies that have evaluated the specificities of the work of an anesthesiologist and the typical situations produced by the current COVID-19 pandemic. We have added some of these studies to the discussion. The second paragraph of the Discussion now is as follows:
“During epidemics, front-line anaesthetists are among the most vulnerable HCWs on account of infections and mental health problems [29-32]. All the effects observed in our sample have been reported by other cross-sectional studies on HCWs engaged on the frontline during the pandemic”
In the chapter on "Participants", it is mentioned that the sample was composed of 120 anesthesiologists. However, as the authors state "The cohort varied since many workers who had participated in the baseline left the hospital in the course of the year". The authors add "The percentage of trainees in the cohort increased significantly during our survey because the hospital hired them under fixed-term contracts to meet the care needs posed by the pandemic". Despite this, it is not explicitly stated which a number of hired trainees integrated the sample. This is of total relevance considering the nature of the concepts and variables under study. Let's see: assuming that hired trainees were not previously part of the hospital, they will need an adequate work environment to develop their activity. If they do not find such an environment, particularly in a situation such as the one underlying the various waves of the pandemic, they will be particularly exposed both from a professional and personal point of view. Therefore, it is essential to understand the relative weight of hired trainees in the global sample, and if there are differences in the results according to the length of service and the fact of being a hired trainee. This is particularly relevant because, as the authors state "The excessive workload could be remedied by increasing staff, but adequately trained personnel are not available and, as we have seen, the hiring of young physicians leads to training problems".
R.: The theme posed by the reviewer is very important. We can reassure him/her that all residents were already at work in the hospital before the outbreak of the pandemic. In Italy, the specialization in Anaesthesiology and Intensive Care has a duration of 4 years. All trainees attend the hospital and carry out care activities under the responsibility of a tutor. At the beginning of the pandemic, the hospital hired fourth-year trainees (who were already working in the university hospital) on fixed-term contracts, assigning them autonomous care tasks in the COVID-19 hub-centre. During the pandemic, some third-year postgraduates were also hired. All residents were part of the hospital, their responsibilities and number of contacts with COVID-19 patients only increased during the pandemic. We have reported the number and percentage of trainees who participated in the baseline survey and in the other surveys in Table 1, second line. To better clarify the fact that the trainees were already working, we added a sentence. The text is now as follows:
“The percentage of trainees in the cohort increased significantly during our survey because the hospital hired them under fixed-term contracts, moving them from the general hospital where they served at the COVID-19 centre, to meet the care needs posed by the pandemic.”
As reported above, in a previous study, which has been accepted for publication in Industrial Health [ref.54], we compared the responses of trainees and other anaesthetists. The only significant difference was the level of informational justice, which was significantly lower in trainees than in older employees, although both groups had the same training. These are precisely the training problems to which we intend to refer readers who will have an interest in reading this article. We added this reference in the manuscript.
Finally, table 2 shows some results that need some additional explanation. Let us see: at the same time that the workload increased and the work became more repetitive and monotonous, isolation at work decreased slightly and isolation in life decreased significantly; however, the factors that can contribute to increasing resilience such as the time devoted to physical activity, meditation, prayer or spiritual activities were reduced or greatly reduced in most workers; conclusion: there seems to be some inconsistency in these data or else there are other variables that will help to better understand them.
Response: The reviewer must consider that those reported are the perceptions of the workers and that therefore they have the typical latency of all perceptions. We have added a statement at the beginning of the results, to remind readers that those obtained from the questionnaires are subjective data. Respondents indicated a workload that was increasing from the first to the second and to the third survey. Objective data on the number of hospitalizations do not correspond to this data. What workers report, therefore, is their subjective perception of workloads. It is absolutely plausible that a person considers the workload the greater the longer the request is prolonged. Hence, it is understandable that 50% of respondents reported increased/ greatly increased workload at the baseline, and this share increased, while hospital records report a stationary number of patients.
Likewise, some workers will immediately notice that the work always takes place on patients with the same diagnosis and therapy, while many others will take a few months for this to be noticed or bothered. Some workers already report at the baseline a more frequent need to inform of the death of a relative, while others perceive this problem only later, even if the mortality rate remains stationary.
Even the little time left for meditation is something that is gradually felt because no one complains if one day he has had too much to do, but if it lasts too long it becomes very evident.
Regarding physical activity, an important contribution to recovery was made by the end of the ban on going to gyms, just as social activities resumed when it was possible to relax the lockdown measures. However, it is not certain that all workers were able to take advantage of these opportunities, because some of them were prevented from doing so by too much work. The data expressed in the table are the sum of all the different situations and personal experiences and they should be interpreted in this way.
Round 2
Reviewer 2 Report
The modifications made by the authors are adequate to publish the article.
Reviewer 3 Report
All my recommendations have been answered
Reviewer 4 Report
I am satisfied with the amendments and consider that the authors have responded adequately to my considerations.